# Assessment Method Integrating Visibility and Toxic Gas for Road Tunnel Fires Using 2D Maps for Identifying Risks in the Smoke Environment

**Huei-Ru Hsieh [1], Hung-Chieh Chung [1,2,*], Nobuyoshi Kawabata [3], Miho Seike [4], Masato Hasegawa [5], Shen-Wen Chien [6] and Tzu-Sheng Shen [6]**

1 Division of Mechanical Science and Engineering, Graduate School of Natural Science and Technology, Kanazawa University, Kakuma-machi, Kanazawa 9201164, Ishikawa, Japan

2 Fire Prevention Division, Kaohsiung City Fire Bureau, Kaohsiung City 80670, Taiwan

3 Faculty of Production Systems Engineering and Sciences, Komatsu University, Nu 1-3, Shicho-machi, Komatsu 9238511, Ishikawa, Japan

4 Graduate School of Advanced Science and Engineering, Transdisciplinary Science and Engineering Program, Hiroshima University, 1-5-1 Kagamiyama, Higashi-Hiroshima 7398529, Hiroshima, Japan

5 Department of Mechanical Engineering, National Institute of Technology, Ishikawa College, Kitachujo, Tsubata 9290392, Ishikawa, Japan

6 Department of Fire Science, Central Police University, Taoyuan City 33304, Taiwan

* Correspondence: fc751126@kcg.gov.tw; Tel.: +886-953662519

**Abstract:** This study proposes an assessment method to quantify the risks of the smoke environment for road tunnel fire safety based on previous studies. The assessment method integrates visibility and toxic gases to address the hazards of smoke distribution more comprehensively. Considering that the hazards of visibility reduction and toxic gases for tunnel users vary with exposure time and location in a fire event, the smoke environment (SE) levels are defined as a function of longitudinal location and time. The SE levels simplify smoke distribution as calculated from 3D computational fluid dynamics (CFDs). For easily identifying SE risks, SE levels are illustrated on a 2D map to analyze the potential hazard by quantifying specific areas and times of smoke exposure. To demonstrate the applicability of the assessment method of this study, cases are carried out using CFD simulation to investigate the risks associated with tunnel fires with various tunnel cross-section types, longitudinal velocities, and gradients. In the analysis of the SE level in different cross-section types and longitudinal velocities under the condition of no vehicle, a velocity of 0.9–1.1 m/s can maintain a less serious SE level both upstream and downstream in a horizontal rectangular tunnel, and 0.3–0.5 m/s in a horizontal horseshoe-shaped tunnel. Both rectangular and horseshoe-shaped tunnels reveal an obvious rise within 10–15 min. In the case of inclined tunnels, for both rectangular and horseshoe-shaped tunnels, the SE level near the fire source obviously deteriorates. Thus, the longitudinal velocity range for the purpose of maintaining a relatively less serious SE level should be slightly reduced for inclined tunnels compared with horizontal tunnels.

**Keywords:** quantitative assessment; extinction coefficient; toxic gas; SE levels; computational fluid dynamics (CFDs)

## 1. Introduction

To assess the fire safety road tunnels, risk analysis as a tool is widely used [1,2], and quantitative risk analysis (QRA) approaches have been investigated by several researchers [3–19].

Kohl et al. (2006) [5] and Kohl and Forster (2012) [6] focused on an integrated quantitative risk analysis consisting of two components, incident frequencies and consequence values, for defined scenarios to calculate the risk value in the event tree. For investigating the risk acceptance criteria of road tunnels, Benekos and Diamantidis (2017) [7] pointed

out that the semi-quantitative risk matrix classification approach and the quantitative risk assessment model (QRAM) can quickly screen and analyze the risk in road tunnels but can offer only preliminary insight. The authors suggested that a quantitative analysis can be used for a specific hazard investigation; however, a more careful investigation of the underlying factors that drive risks is needed. Ntzeremes and Kirytopoulos (2018) [8] proposed a SIREN method based a stochastic-based approach that considered the tunnel characteristics and variability and uncertainty of stochastic parameters of the tunnel system. Qin and Kang (2019) [9] applied a fuzzy analytic hierarchy process, combining qualitative and quantitative analysis to propose a judgment matrix for their road tunnel fire risk index. Caliendo and Genovese (2021) [10] investigated the potential consequences of dangerous goods vehicle accidents, including the transport of liquid hydrogen, using DG-QRAM software to analyze the possibility of investigating two different risk scenarios in unidirectional tunnels. Quantitative risk criteria for tunnel transportation in the Republic of Slovenia were proposed by Vidar (2022) [11], who used CFD analysis to consider soot density and temperature to evaluate fatalities during fire development. Haddad and Harun (2023) [12] presented a new QRA model for UK road tunnels using the event tree approach and road tunnel statistical data in the UK for the quantitative analysis of frequency and consequences.

Generally, the computational fluid dynamics (CFDs) model is considered a useful and widely adopted tool for QRA in modeling smoke behavior in tunnel fires. Evacuation models can be compared based on pre-determined acceptable criteria regarding quantitative consequences, including the potential number of fatalities and likelihood of failure to self-evacuate. The results of such a QRA are highly related to both the smoke distribution and the time of exposure.

CFD technical is a practical approach to estimating the distribution of smoke over time. When applied in the QRA, the determination of the hazard level mainly depends on the objectives to be achieved (such as the determination of the number of people that failed to self-evacuate or the number of fatalities) or the adopted risk acceptance criteria based on minimum safety requirements.

From a review of the literature, it is apparent that the quantification of the potential hazard of smoke distribution to tunnel users can be divided into two main aspects [13]. One is the influence of visibility on walking, and the other is incapacitation or death due to the accumulated toxic dose in a given time duration. Purser (1989 and 2016) [14,15] proposed a toxic gas-based hazard calculation model by considering the concept of fractional effective dose (FED). Purser (2009) [16] applied CFD and FED modeling to investigate the Mont Blanc Tunnel fire, revealed that tunnel environments deteriorate seriously after 6 min due to rapid loss of visibility, increase in CO concentration, and rise in temperature and heat flux. Qu et al. (2013) [17] focused on the fire consequence regarding the number of casualties caused by toxic gases. The proposed risk analysis approach combines the temperatures and toxic gas concentrations ($CO$, $CO_2$, and $O_2$) estimated by the Fire Dynamics Simulator (FDS) and fractional effective dose (FED) model to estimate fatality rates at different locations in given periods. Seike et al. (2017) [18], in their consequence analysis, used the extinction coefficient ($C_s$) as the hazard indicator and proposed a quantitative method for evaluating the influence of visibility on people that are evacuated from an in-smoke environment. Huang et al. (2021) [19] compared two assessment approaches based on the extinction coefficient index (termed Japanese-style assessment) and CO concentration (termed European-style assessment). The study found that the CO concentration-based assessment approaches reflected relatively optimistic results (in the same fire scenario: 30 MW HRR; longitudinal ventilation velocity set at 0 m/s). This implies that visibility-based assessment for evacuation safety would be relatively strict because the basis of assessment focuses on whether self-evacuation is feasible. Although this comparison aimed at establishing which of these two assessment approaches was more reasonable, no definitive conclusion was reached.

Considering the above, to discuss the hazards of smoke distribution more comprehensively, it is necessary to establish appropriate classifications for smoke hazard levels; therefore, this study aims to propose a quantitative assessment method for smoke hazards that integrates toxic gases (CO and HCN) and visibility to define smoke hazard levels. We used 2D images to analyze the potential hazard by quantifying specific areas and times of smoke exposure. The fire simulation tool for modeling smoke distribution, tunnel geometry setting, and simulation conditions is explained in Section 2. The definition of smoke hazards using smoke environment (SE) levels are provided in Section 3. Section 4 includes case simulations with various longitudinal velocities, cross-section types, and gradients to further discuss the applicability of the SE levels. The main findings are summarized in Section 5.

## 2. CFD Simulation for Smoke Behavior

In this study, we use the 3D CFD code (Fireles) developed by Kawabata et al. (1998) [20] to conduct the simulation. Fireles was developed based on large eddy simulation (LES) model and used the Smagorinsky model to simulate the turbulence. The main governing equations of Fireles include the conservation of energy equation, the momentum equation, the continuity equation, the equation of state, and the equation of smoke concentration. Other mathematical equations and the boundary condition of the fields of velocity and temperature in the present CFD model were reported in the study of Tung et al., 2023 [21]. The above equations are solved by the finite volume method. Regarding the spatial schemes, the fourth-order central-difference scheme is applied to the momentum equation, and third-order and first-order upwind-difference scheme is applied to energy and smoke concentration equations. The second-order central-difference scheme is applied to other mathematical equations in our CFD model for spatial differentials. The results (such as smoke and toxic gas concentration, velocity, temperature, and airflow direction) are obtained through discretizing of the tunnel volume based on the grid division.

In previous studies, the simulation capabilities of Fireles based on the comparison of the numerical simulations and experimental results in both model-scale and full-scale tunnels were validated. Moreover, Fireles has been used as a simulation tool for tunnel fire safety assessment and adopted in Japan.

Since the simulation results are related to the grid division, we also conducted grid independence analysis before the formal simulation, as specified in Section 2.1. Moreover, to ensure the capability of turbulence simulation, a further examination of the turbulence state was conducted through using the friction factor and the turbulence intensity, as detailed in Section 2.2.

### 2.1. Grid Independence Test

To obtain accurate results without generating difference in the numerical analysis based on various grid conditions, the grid independence test was performed to determine a suitable grid. The indicator of back-layering length was considered to assess the CFD results in this study, as it is a critical parameter that is representative of smoke behavior in tunnel fires. Two sets of grid independence tests were conducted by considering two cross-section shapes (rectangle shape and horseshoe shape). The simulation space was 1000 m (L) × 10 m (W) × 5 m (H) for the rectangular tunnel and 1000 m (L) × 11 m (W) × 6.8 m (H), with one portal closed, for the horseshoe-shaped tunnel.

The average longitudinal ventilation velocity (U) was 0 m/s. In the rectangular tunnel, the heat release rate (HRR) was designated to reach 10 MW at 30 s and then maintain a steady state. In the horseshoe-shaped tunnel, the HRR curve was the same as the rectangular tunnel, but the max HRR was 10 MW (corresponding to the 20 MW convective HRR under the condition of both side portals being open). The simulation time was from the start of heat generation until 300 s elapsed.

In this study, five different grid divisions were investigated to examine the grid independence according to the comparisons between the simulation results. The details

of the differences between the five grids are shown in Table 1. The total number of grids includes the grids division of the simulation space as well as the grid division of boundaries.

**Table 1.** Specifications of grids.

| Rectangular Tunnel | | | | | |
|---|---|---|---|---|---|
| | Grid 0 | Grid 1 | Grid 2 | Grid 3 | Grid 4 |
| Number of grids in the three directions ($M_x$, $M_y$, $M_z$) | 3000, 43, 27 | 2850, 41, 25 | 2720, 39, 23 | 2300, 33, 21 | 1400, 29, 17 |
| $\Delta x$, $\Delta y$, $\Delta z$ (m) $\sqrt[3]{\Delta x \Delta y \Delta z}$ (m) | 0.333, 0.233, 0.185 0.243 | 0.351, 0.244, 0.200 0.258 | 0.368, 0.256, 0.217 0.274 | 0.435, 0.303, 0.238 0.315 | 0.714, 0.345, 0.294 0.417 |
| Total number of grids (including simulation section and boundary section) | 3,904,101 | 3,298,419 | 2,777,969 | 1,872,949 | 819,401 |
| Horseshoe-shaped tunnel | | | | | |
| | Grid 0 | Grid 1 | Grid 2 | Grid 3 | Grid 4 |
| Number of grids in the three directions ($M_x$, $M_y$, $M_z$) | 3000, 45, 35 | 2800, 41, 33 | 2400, 37, 29 | 2200, 35, 27 | 1400, 31, 21 |
| $\Delta x$, $\Delta y$, $\Delta z$ (m) $\sqrt[3]{\Delta x \Delta y \Delta z}$ (m) | 0.333, 0.244, 0.194 0.251 | 0.357, 0.268, 0.206 0.270 | 0.416, 0.297, 0.234 0.307 | 0.455, 0.314, 0.252 0.330 | 0.714, 0.355, 0.324 0.435 |
| Total number of grids (including simulation section and boundary section) | 4,396,145 | 3,542,969 | 2,441,553 | 1,987,255 | 900,635 |

We adopted the parameter of back-layering length to examine grid independence. The deviation of the average back-layering length (in the simulation time of 285–300 s) between the five grid divisions was compared with the basis of the back-layering length of Grid 0 (the equation can be express as: $|L_b - L_{bGrid0}|/L_{bGrid0}$; see Figure 1). The grid size near Grid 0 would have a smaller deviation. In the rectangular tunnel, the deviations of Grid 1, Grid 2, Grid 3, and Grid 4 were 0.3%, 0.3%, 0.7%, and 2.5%, respectively. In the horseshoe-shaped tunnel, the deviations of Grid 1, Grid 2, Grid 3, and Grid 4 were 0.9%, 1.7%, 1.7%, and 7.8%, respectively. A relatively significant deviation from the rectangle shape was observed, but deviation in both tunnel shapes was at an acceptable level in grid sizes smaller than Grid 3 (an error of no more than 5%).

Moreover, we consider that the deviation of back-layering length in different grid divisions is due not only to the difference in grid resolution but to the fire source reproducibility in different grid sizes. When the fire source is modeled by different grid divisions, the area and shape of the fire source have minor differences due to the change in grid size. However, even if this factor is considered, we can find that the deviation in the results for Grid 1 and Grid 0 in these two tunnel types is less than 1%, indicating that the effect of grid size is small. Based on the above analysis, we selected Grid 0 in both the rectangle-shaped and horseshoe-shaped tunnels for the simulation to ensure sufficient accuracy in simulating smoke descending behavior.

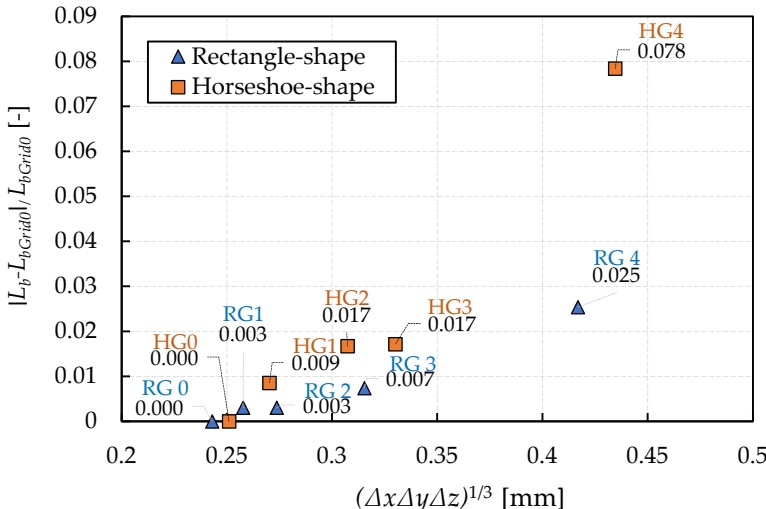

**Figure 1.** Deviation of smoke back-layering length in five grid sizes.

### 2.2. Analysis of Smagorinsky Coefficient for Turbulence Reproducibility

When simulating the dynamics of hot gas flow and smoke using the CFD model, examination of the simulation condition, constant of the model, grid division, and so on, is necessary for appropriately capturing the true behavior of the turbulence phenomenon. The constant Smagorinsky coefficient ($C_{sgs}$) is a critical parameter to reflect the energy dissipation from large scale to small scale, depending on the flow types; it is associated with the predictive ability of the LES model.

The constant $C_{sgs}$ of 0.20 is generally adopted as a simulation parameter in tunnel fires. The fire region is the driving force for the changes in the forced flow conditions [22] when applying FDS 4.0 [23] for simulation. However, in theory, a greater Smagorinsky coefficient would lead to the loss of the dynamic of large-scale eddies by causing the reduction in the generation of large eddies (eddy size over the filter) because the turbulent kinematic viscosity coefficient of small-scale eddies would increase.

On the other hand, in FDS 4.0, the convective terms in the momentum conservation equation are approximated by second-order finite differences and applied to the upwind-biased differences scheme in the predictor step and downwind-biased difference scheme in the corrector step [23]. In the study of Kawabata et al. (2003) [24], a Smagorinsky coefficient of 0.125–0.15 was reported to be suitable by considering the convective terms in the momentum conservation equation. The momentum conservation equation is approximated by the fourth-order central-difference scheme, regardless of whether there is a no-fire or fire condition in the tunnel space. We consider that the usage of the above numerical calculation results in the difference of Smagorinsky coefficient for tunnel fire modeling in FDS 4.0 or other CFD models.

Considering the sensitivity of $C_{sgs}$ for turbulence modeling and the fourth-order central-difference approach, Tung et al. (2023) examined turbulence simulation through the adjustment of $C_{sgs}$ in their study [21]. They reported that $C_{sgs}$ set at smaller than 0.15 would result in relatively good reproducibility in turbulence simulation for horseshoe-shaped tunnels.

We considered a reasonable $C_{sgs}$ value as smaller than 0.15 in accordance with Tung et al., 2023 [21], because our simulated horseshoe-shaped tunnels were similar to the one used in their study. Moreover, since there is a minor difference in grid division size between the rectangular and the horseshoe-shaped tunnels, we selected a $C_{sgs}$ of 0.13 for both rectangular and horseshoe-shaped tunnels in the present simulation. Table 2 lists the other calculation settings of the present CFD code.

**Table 2.** Calculation setting of the simulator.

| **Boundary Conditions** | | |
| --- | --- | --- |
| The surface of a wall | Velocity | Equations (A20)–(A22) in Appendix from Tung et al., 2023 [21] |
| | Temperature | Heat transfer coefficient (Jürges, 1924) [25] |
| | Heat conduction in the wall | 1D heat-conduction equation |
| | $+x$ inlet | Uniform wind velocity of $x$ direction |
| | $-x$ outlet | Constant pressure ($p = 0$) |
| **Calculation schemes for convective term** | | |
| | Velocity | Fourth-order central-difference scheme |
| | Temperature | Third-order upwind-difference scheme |
| | Smoke | First-order upwind-difference schemes |
| **Constant terms in the calculation** | | |
| | Courant number | 0.2 |
| | Smagorinsky coefficient | 0.13 |
| | Turbulent Prantl number | 0.7 |
| | Turbulent Schmitt number | 0.7 |

*2.3. Simulation Conditions*

The rectangular tunnel and horseshoe-shaped tunnel were used in the CFD analysis in this study. The rectangular tunnel is 10 m in width and 5 m in height; the horseshoe-shaped tunnel is 11 m in width and 6.8 m in height. The equal grid division region for the simulation space is considered as 1300 m in length and is part of a several-kilometers-long tunnel. However, in the case of the inclined tunnel with 0 m/s (U), we consider that longitudinal gradients would cause the smoke layer to spread unequally on the two sides; thus, we extended the simulation space to 1800 m in this case. The tunnel slopes are varied at 0%, 2%, and 4%.

To reduce the effect of the tunnel openings in the calculations, the calculation range along the $x$-axis is extended by 100 m at each opening to be a boundary section, making the total length of the simulation tunnels 1500 m or 2000 m. Figure 2 shows the schematic views of the simulation tunnels.

At the right opening of the tunnel, ventilation air is supplied to the simulated tunnel. The longitudinal ventilation velocity (U) is set in the range of 0 m/s to 2.2 m/s. The longitudinal velocity is set to reach the target velocity in 30 s. The purpose of this study is to investigate the distributions of toxic gas and the visibility in smoke environments on both sides of the fire source. For a basic analysis, we first examined the case where there are no vehicles involved. More detailed parameters of longitudinal ventilation and gradients for simulation cases are listed in Table 3.

The fire source is set at the origin of the coordinates (x = 0) in the simulation. The simulation length on two sides of the fire source is adjusted depending on velocity and gradient conditions. In order to simplify the conditions and parameters in numerical simulation, the combustion reactions (chemical reactions) are ignored in the present simulations. Relatively, a simplified model based on existing heat release and smoke generation from past experiments is applied. Although the designed fire is different between countries, each country proposed its representative fire scenarios mainly based on the setting of time-related heat release rate and gas generation rate (or soot yield). For example, the representative fire scenario is a bus fire based on actual vehicle fire experiments (school bus fire experiments in EUREKA project [26] and a single large bus fire test in the Shimizu No.3 tunnel in Japan [27]), the max HRR is 30 MW (convective component is around 18 MW to 20 MW), and the smoke generation rate is 90 g/s. Therefore, the total HRR is set 30 MW in this study, which assumes a bus fire in a tunnel based on the representative scenario in tunnel fire risk assessment in Japan.

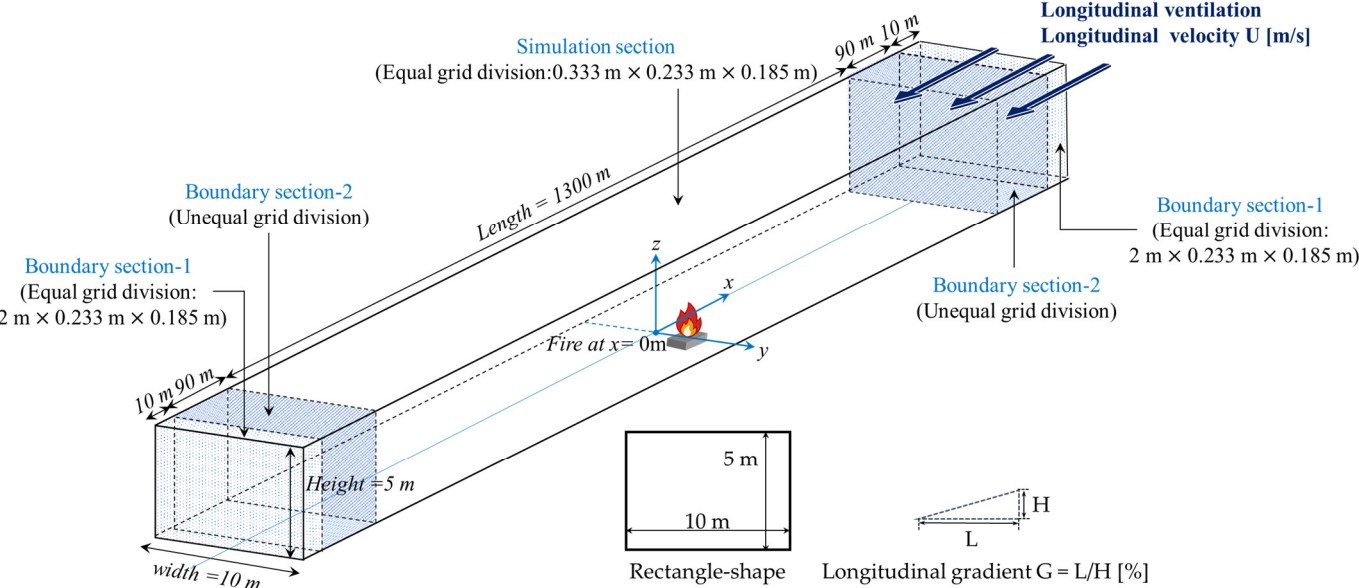

(a) Rectangular tunnel

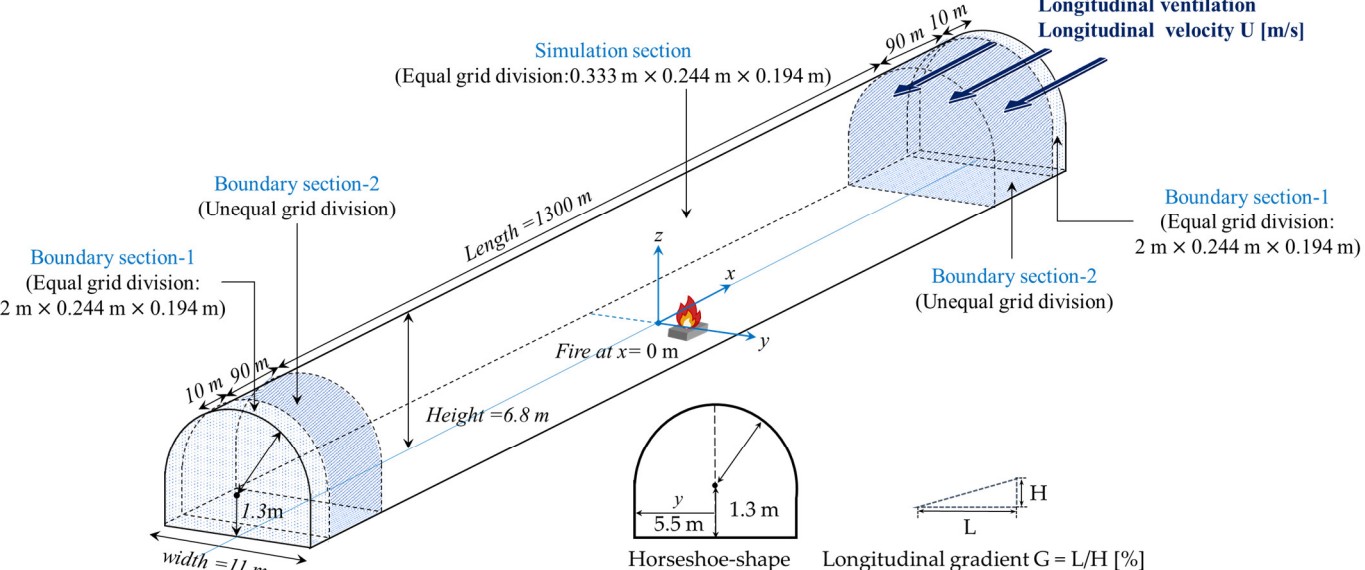

(b) Horseshoe-shaped tunnel

**Figure 2.** Cross-section and schematic view of the simulated tunnel. (**a**) Rectangular tunnel. (**b**) Horseshoe-shaped tunnel.

The heat generated from a fire mainly comprises the convective heat of hot gas and the radiant heat that is absorbed by the smoke and walls. Moreover, the radiation from a fire's heat airflow and smoke particles can ignite materials near the fire. With smoke propagating along the ceiling, heat is absorbed by the ceiling. The heated smoke layers also transfer radiant heat to the smoke particles or toward the surroundings when propagating. This thermal radiation absorption and reflection process also occurs in the smoke layers far away from the fire source; this is a complicated heat exchange phenomenon that is difficult to reproduce in simulations, as it is considered a minor influence on the smoke

distribution compared with the influence of heat convection. In addition, full-scale tunnel fire experiments show convection makes up a large proportion of the total HRR and can be expected to vary between 60 and 70% [26].

**Table 3.** Simulation conditions setting.

| Tunnel | Rectangular Tunnel | Horseshoe-Shaped Tunnel |
|---|---|---|
| Tunnel dimensions (simulation section) | 1300 m (L) × 10 m (W) × 5 m (H) | 1300 m (L) × 11 m (W) × 6.8 m (H) |
| Total number of grids (simulation section) | 5,348,259 | 6,022,267 |
| Grid size (simulation section) | 0.333 m × 0.233 m × 0.185 m | 0.333 m × 0.244 m × 0.194 m |
| Total HRR (convective HRR) | 30 MW (Convective HRR = 20 MW) | |
| Traffic condition | No traffic blockage (no cars near the fire source) | |
| Longitudinal ventilation velocity (U) | 0 m/s, 0.3 m/s, 0.5 m/s, 0.9 m/s, 1.1 m/s, 1.3 m/s, 1.5 m/s, 2.0 m/s, 2.2 m/s | |
| Longitudinal gradients (G) | 0%, 2%, 4% | |
| Simulation time | 900 s | |

Note: The simulation space extends to 1800 m (L) × 11 m (W) × 6.8 m (H) in the case of an inclined tunnel with longitudinal ventilation of 0 m/s; the total number of grids increase to 7,299,799 (rectangular tunnel) and 8,219,747 (horseshoe-shaped tunnel).

As discussed above, the effects of radiant heat mainly occur in the region near a fire; however, our focus is on the smoke distribution away from the fire source. Thus, we do not further model the heat radiation and mainly discuss the influences of convection on smoke propagation in this study. We assume that 67% of the total HRR is convective and estimate the peak HRR to reach 30 MW [27,28].

In the design fire curve shown in Figure 3, the convective HRR and smoke generation rate (SGR) are time-related developments. As the fire develops, the area of heat generation is gradually enlarged and increases with the fire growth coefficient $\alpha = 0.08$ kW/s$^2$ until 480 s, after which it stays constant. The fire growth rate is similar to an ultra-fast t$^2$ fire. The SGR is assumed to be the same as the HRR curve, with the trend of the time square, and reaches the constant of 90 g/s after 488 s.

*2.4. Extinction Coefficient*

The extinction coefficient ($C_s$) is one of the parameters used to evaluate visibility. It is used to characterize particle species and smoke concentration in order to evaluate smoke propagation and visibility in a tunnel. It can be obtained by the following Equation (1) [29]:

$$C_s = -\frac{1}{D}\log_e\left(\frac{I}{I_0}\right) \tag{1}$$

where $I$ is the intensity of light through smoke [cd], $I_0$ is the intensity of the incident light [cd], and $D$ is light path length [m].

$C_s$ is one of the factors to determine the risk level of a smoke environment. It is usually calculated by numerical derivation or obtained as data from experiments. The physical quantities in CFD simulation are derived using a governing equation based on the conservation law, such as that for smoke mass concentration.

In addition, the relationship of smoke mass concentration (M) and $C_s$ has been reported by previous studies, which proposed the linear correlation between M and low $C_s$, with the $C_s$ growth rate decreasing gradually with increasing M at high $C_s$ (above 2 m$^{-1}$) [30,31]. Therefore, we used the mathematical relation between smoke mass concentration and $C_s$ to derive the $C_s$ of this study. The smoke mass concentration (M) is solved by CFD analysis

based on the conservation law; then, the extinction coefficient ($C_s$) can be calculated by the following mathematical relations [32]:

$$C_s = 10M \qquad\qquad M \leq 0.26 \left[g/m^3\right] \qquad\qquad (2)$$

$$C_s = 1.73\ln(M) + 4.94 \qquad M > 0.26 \left[g/m^3\right] \qquad\qquad (3)$$

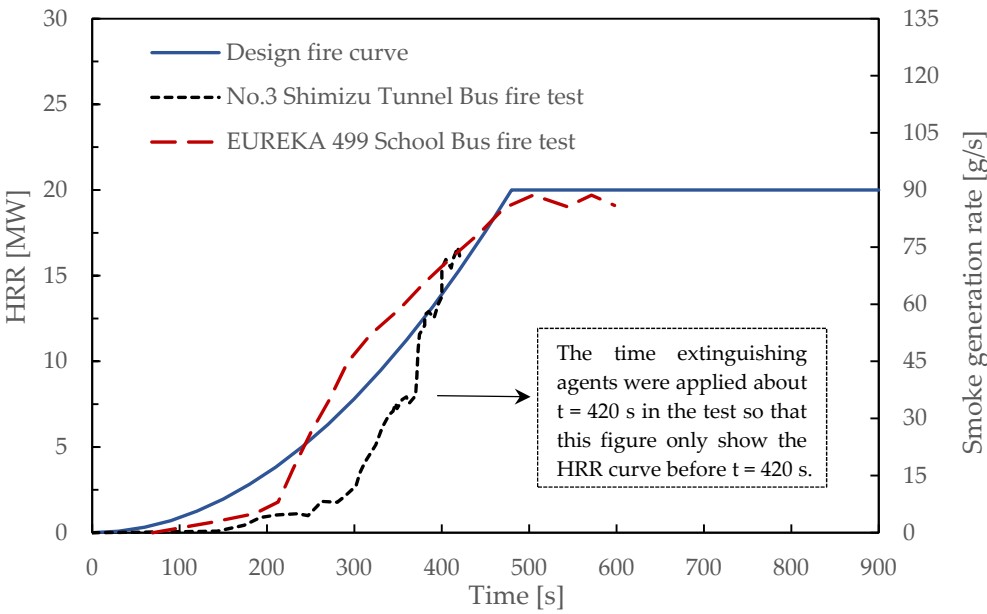

**Figure 3.** Convective HRR and smoke generation rate.

Equation (3) expressed the correlation of the smoke mass concentration (M, g/m$^3$) and extinction coefficient ($C_s$, m$^{-1}$) of transmission of a beam. The derivation of Equation (3), that is divided into two situations: low smoke density (the above equation) and high smoke density (the below equation). In low smoke density, a cloud of soot particles in smoke is approximated as a sphere. The assumption of the projection of each soot does not overlap, the difference ($I_0$-I) between the intensity of the light without smoke ($I_0$) and the intensity of the light through smoke (I) is proportional to nd$^2$ (n is soot particle number, and d is soot particle diameter) [27]. The below equation of Equation (3) is applied to the experimental formula that expresses the correlation between $C_s$ concentration and smoke mass concentration based on the study of Takao et al. (2003) [33]. The above equation ($C_s$ = 10 M) is determined at the threshold value of M = 0.26 g/m$^3$ shown as Equation (3). The below equation is without physical meaning, which is expressed as the relation between the measurement values of smoke mass concentration and $C_s$.

### 2.5. Toxic Gas Generation Rate

The smoke generation rate is proportional to the time-related HRR curve which is the hypothesis of this study. The smoke generation rate reaches 90 g/s at 480 s, and thereafter maintains a constant rate. The constant value of 90 g/s is taken from a bus fire test in a real-scale tunnel [32], which measured a max smoke generation rate of 95.6 g/s for a 33 MW fire scale (with a soot yield ratio corresponding to 0.127 g/g); in addition, 90 g/s for a 30 MW fire scale is used in the quantitative road tunnel risk analysis tool TuRisMo as the reference [34].

Since CO and HCN are the major asphyxiant gases causing incapacitation and death in fires [35], these two toxic gases are considered in this study as causes of environmental risk in tunnel fires. In the literature that we explored, there were no detailed experiments

and measures of CO generation rate and HCN generation rate aimed at representative or simplified settings for toxic gases. The specific setting of 108 vg/s and 27 g/s were preliminary used as the max CO generation rate and HCN generation rate in this study, and are the parameters set in the quantitative road tunnel risk and analysis tool TuRisMo (Austrian tunnel risk model). In addition, the growth curves with time of the generation of these substances are proportional to the fire growth rate in TuRisMo [34].

Therefore, the CO and HCN generation rates are assumed to be proportional to the fire growth rate and smoke generation rate over 480 s in this study, after which they maintain a constant rate. In fire scenarios with 30 MW and longitudinal ventilation, the constant CO and HCN generation rates are assumed to be 108 g/s and 27 g/s.

## 3. Smoke Environment (SE) Map Integrating Visibility and Toxic Gas

### 3.1. Smoke Exposure Risk and Corresponding Smoke Environment Levels

Heat, smoke, and toxic gas are the main causes of casualties in a tunnel fire. Additionally, worsening visibility can indirectly lead to death [29]. In the early stages of fires, people trapped in tunnels can miss emergency exits because the visibility decreases rapidly; the inhalation of toxic gas during delayed evacuation attempts can result in confusion, loss of consciousness, and eventually death due to hypoxia.

To evaluate the influences of visibility and toxic gas on evacuees in tunnels, we modified the visibility-based SE map proposed by Seike et al. (2017) [18]. We integrated the two risk indexes of visibility and toxic gas to define SE levels and used a graphical approach to describe the smoke distribution. The risk in the smoke environment is assessed through predefined "risk levels", which reflect the range of smoke diffusion and the evacuation tenability when evacuees are exposed to toxic gases or dense smoke.

In this study, SE levels are a function of $x$ and $t$; $x$ represents smoke arriving at each longitudinal location ($x$), and $t$ is time in minutes. The SE levels are defined based on the height of smoke layer, extinction coefficient ($C_s$), and smoke toxicity value (STV) divided into eight levels (Levels 0–7). The smoke propagation in tunnel fires is visualized using a 2D map.

As explained earlier, visibility is the first threat from smoke for evacuees during the evacuation phase; they can be trapped in a dangerous smoke environment full of toxic gases if they cannot evacuate quickly.

Therefore, we considered the relationship between smoke propagation and time. The SE levels 0–3 in Table 4 define the risk of smoke by visibility. In this study, a smoke toxicity value (STV) is defined to express the risks of a smoke environment and serves as an index for SE levels 4–7, which represent the danger of toxic gases CO and HCN in fires. The SE levels are summarized in Table 4 and further detailed in the following section.

Figure 4 is a schematic diagram that illustrates the transformation from 3D CFD simulation results to a 2D SE map (from the picture on the right side to the picture on the left side). As seen in the right picture of Figure 4, the smoke distribution ($C_s$ value) in a tunnel fire can be shown using 2D images in the longitudinal direction ($x$) and vertical direction ($z$). The side view of smoke distribution can be exported depending on the simulation time. To express the hazard from smoke, the SE levels are displayed on the 2D map for different longitudinal locations (shown as $x$) and time (shown as $t$). Plot colors are used to distinguish the SE levels on the left image of Figure 4.

### 3.2. Classification of SE Level Considering Visibility

SE levels 0–3 in Table 4 are graded by considering the extinction coefficient, which reflects the smoke propagation and visual distraction to tunnel users. An SE level of 0 (in white) indicates that $C_s > 0.4 \text{ m}^{-1}$ at the height above 4.5 m in the rectangle-shaped and 6.0 m in the horseshoe-shaped tunnel. The smoke layer is at around 88–90% of tunnel height, even if in different cross-section types. $C_s > 0.4 \text{ m}^{-1}$ indicates dense smoke, which disrupts human visibility. Further, it affects walking speed, which is usually considered in the safety criteria of Japan's road tunnel fire risk assessment. In addition, visibility distance

is usually set at a threshold value of 10 m, a value widely used as the acceptable tunnel-fire safety index [13]. According to the function of the visibility of reflective signs at the obscuration threshold, visibility $[m] = \frac{2}{C_s} \sim \frac{4}{C_s}$ [29], the extinction coefficient ($C_s$) for the visibility distance of 10 m is $0.2\ m^{-1}$–$0.4\ m^{-1}$. This shows that the criteria for evaluating the risk of smoke are similar when using the extinction coefficient and the visibility distance.

**Table 4.** Visualization of Smoke risk levels.

| Risk Level | | $C_s$ [m⁻¹] | STV | Height Z [m] | Toxic Gas | |
|---|---|---|---|---|---|---|
| | | | | | CO [ppm] | HCN [ppm] |
| 0 | | | | | | |
| | | 0.4 | | 4.5/6.0 (Rectangular / Horseshoe-shaped) | | |
| 1 | Blue | | | | | |
| | | 0.4 | | 3.0/4.0 (Rectangular / Horseshoe-shaped) | | |
| 2 | Light blue | | | | | |
| | | 0.4 | | 1.5 | | |
| 3 | Orange | | | | | |
| | | | 0.15 | 1.5 | 84 | 21 |
| 4 | Green | | | | | |
| | | | 0.30 | 1.5 | 169 | 42 |
| 5 | Purple | | | | | |
| | | | 0.60 | 1.5 | 337 | 84 |
| 6 | Yellow | | | | | |
| | | | 1.00 | 1.5 | 562 | 140 |
| 7 | Red | | | | | |

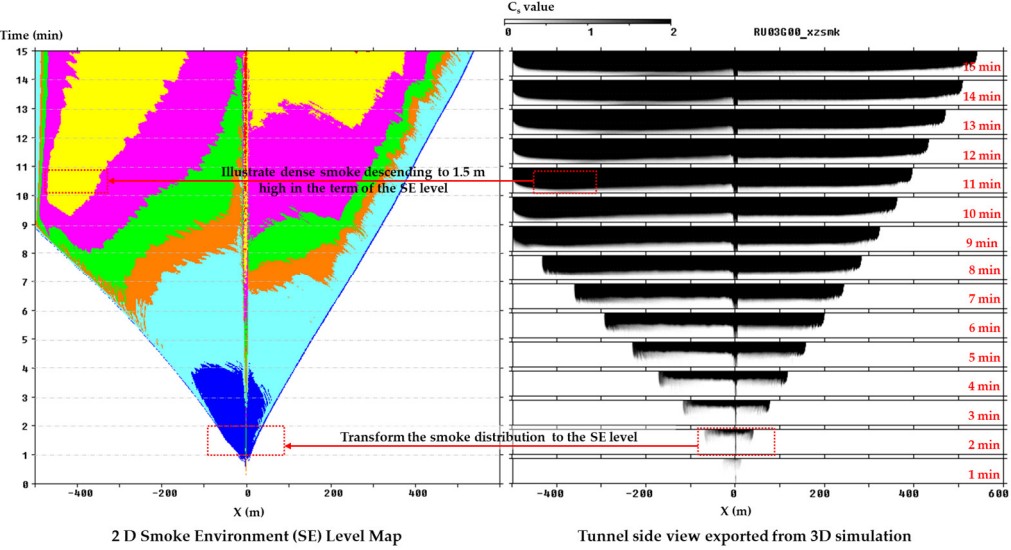

**Figure 4.** Schematic diagram of 2D SE level map as derived from 3D simulation results.

SE level 1 (in blue) illustrates the forefront of smoke propagated along the tunnel ceiling; the reference height is set at 4.5 m (6.0 m in horseshoe shape) and $C_s > 0.4\ m^{-1}$.

SE level 2 (in light blue) indicates smoke diffusion moving toward the road surface; the reference height is set at 3.0 m in the rectangle-shaped tunnel and 4.0 m in the horseshoe-shaped tunnel (around 58–60% of the tunnel height).

SE level 3 (in orange) is graded by considering that people's walking speed would be affected by exposure to smoke. According to Japanese assessment criteria, at 1.5 m of

reference height, an extinction coefficient ($C_s$) exceeding 0.4 m$^{-1}$ is judged as an unsafe evacuation environment [19]; this means that SE level 3 and above in Table 4 represents a dangerous environment for evacuees in Japan.

*3.3. Classification of SE Level Considering Survival*

SE levels 4–7 in Table 4 are graded by considering the influence of toxic gas inhalation. The grade of these levels focuses on the environmental risk that might cause injury and incapacitation due to toxic gases, rather than a reduction in walking speed.

The present toxic gas-based analysis model applies the concept of incapacitation or lethality for the population for a period of exposure. Interactions between individual toxic gases are additive; therefore, smoke toxicity can be estimated by summing the influence of individual toxic components [36]. The influences of toxic components are expressed as the concentration as their fraction of the lethal or incapacitation concentrations. A widely used example is the calculation of the fractional effective dose (FED).

The FED is a practical method that is used to determine whether smoke is life threatening [37–39]. It measures the dose of toxic products people have inhaled during the exposure time to predict the toxic effects on their escape capability [14,15]. In the FDS + Evac model, the gas phase concentrations of $O_2$, $CO_2$, and CO are taken from the FED proposed by the study of Purser as the default concentrations, which represent human incapacitation [40]. The Pathfinder evacuation model considers the effects of $CO_2$ and CO [41]. CO and HCN are discussed in this study because they are the major toxic gases leading to incapacitation and death in fires.

In comparison with the FED, which mainly considers the dose of toxic products absorbed by humans, the smoke toxicity value (STV), in addition to the concept of FED, is developed in this study, as it focuses on the quantitative toxicity risk of a smoke environment. Furthermore, the FED considers the dose of toxic products and duration of the exposure, while the STV mainly considers varieties of instantaneous concentrations of toxic gas but does not include the effect of the duration of the exposure.

The STV can be calculated by the instantaneous concentration of toxic gas divided by the concentration causing lethality or incapacitation within 30 min, as presented in Equation (4). The numerator of STV is the instantaneous toxic gas concentration corresponding to the simulation time. The denominator of STV is effective toxic gas concentration to cause lethality or incapacitation in 30 min. The overall FED received by evacuees can be estimated by integrating the STV over a continuous period of 30 min (1800 s). Thus, FED can be calculated as shown in Equation (5).

$$STV = \frac{[\text{Toxic gas generated concentration}]}{\text{Lethality or incapacitation of toxic gas concentration for 30 min}} \qquad (4)$$

$$SFED = \frac{1}{1800} \int_0^{1800} STV \cdot dt \qquad (5)$$

Experimental data to produce lethality in 50% (LC50) of test animals for 30 min exposures were used in this study. Based on the toxicity experiments in rats, the LC50 values of carbon monoxide (CO) is 4600 ppm for 30 min exposures and the 30 min LC50 values of hydrogen cyanide (HCN) is 160 ppm [42,43]. Therefore, Equation (4) can be expressed as follows:

$$STV = \frac{[CO]}{4600 \text{ ppm}} + \frac{[HCN]}{160 \text{ ppm}} \qquad (6)$$

STV > 1 means that evacuees are exposed to the highest risk environment, which may result in fatality. In addition, since no valid data have been published regarding toxicity when CO concentration and HCN concentration are mixed, data were determined using the time and concentration until death of CO alone and HCN alone.

SE level 7 (in red) is set at STV > 1, the highest level. It should be noted here that the degree of harm reflected in health effects, impaired ability, and fatality is significantly

different; thus, we assume the highest level reflects high risk leading to death. SE levels 4 to 6 are based on the STV to reflect different degrees of environmental risk to health.

SE level 4 (in green) is set at STV > 0.15 which indicates that the toxicity of a smoke environment is less serious, because the exposure time and toxic gas concentration are relatively low and might lead to slight effects, weakness, and minor injuries for evacuees. The SE level 5 is set at STV > 0.3, which indicates that smoke environment might put evacuees at risk of minor injuries or the impaired ability to escape when they stay over 30 min; this is similar to 30 min AEGL-2 values for carbon monoxide [44]. SE level 6 (in yellow) is set at STV > 0.6, which indicates that the smoke environment might lead to serious injuries or life-threatening adverse health effects; this is similar to 60 min AEGL-3 values for carbon monoxide. Therefore, SE levels 4 to 7 can be regarded as reflecting situations where external assistance may be required.

## 4. SE Map Analysis Results

Since the analysis of this study aimed to consider the traffic congestion scenario, the impact of smoke distribution on both the upstream and downstream evacuation environment should be considered. However, if we considered the influence of both vehicles and velocity on smoke diffusion, it would be difficult to distinguish the significance of these two factors. Thus, to simplify the variables that affect the smoke distribution, we chose to analyze the situation without traffic blockage and with no vehicles near the fire source.

### 4.1. SE Map with Various Longitudinal Velocities and Cross-Section Types

Figure 5 illustrates the SE map of variant longitudinal ventilation velocity (U, m/s) in a rectangular tunnel with no inclination. The U is set at 0 m/s–2.2 m/s. The displayed area in Figure 5 is adjusted to reveal the smoke distribution as much as possible under different velocity conditions; the total illustration region is 1000 m (total simulation space is 1800 m). According to Figure 2, the inlet direction of longitudinal ventilation flow is from the right to left side of the tunnel. Therefore, to consider the inlet direction of longitudinal ventilation flow, the right side of the fire is defined as upstream in Figure 5 and the left side of the fire is downstream.

In the environment upstream of the fire, the SE map shows lower longitudinal ventilation velocities and a worse evacuation environment. From the SE map of the rectangular tunnel (Figure 5), we can see that there is little smoke upstream when the U is greater than 2 m/s. When U reaches 2.2 m/s, the approximate back-layering length is shorter than 20 m in the simulation time of 900 s. Thus, it can be estimated that a critical velocity for the rectangular tunnel with a fire scale of 30 MW would be slightly greater than 2.2 m/s. Using the critical velocity equations proposed by Oka and Atkinson (1995) [45], Wu and Bakar (2000) [46], and Li and Ingason (2017) [47], the calculated velocities are 2.45, 2.57, and 3.01 m/s, respectively. The present estimation is relatively close to the equations proposed by Oka and Atkinson (1995) [45] and Wu and Bakar (2000) [46] equation rather than the equation of Li and Ingason (2017) [47].

When U < 1.1 m/s, SE level 3 (in orange) begins to spread to the upstream side after 10 min, and SE level 4 (in green) begins to spread after 12 min. From U ≤ 0.5 m/s, SE level 5 (in purple) begins to spread to the upstream side after 10 min. When U = 0.5 m/s, SE level 6 (in yellow) rapidly spreads to x = 300 m upstream at t = 14 min. When U = 0.3 m/s, SE level 6 (in yellow) occurs earlier (in 12 min) and spreads to x = 400 m upstream at t = 15 min. When U = 0 m/s, SE level 7 (in red) occurs at the location of x = 350 m upstream at t = 13 min and then spreads to x = 300–500 m in 15 min. The tendency of upstream smoke to descend toward the road surface starts away from the fire point and spreads gradually. When U = 0 m/s and 0.3 m/s, it can be found that SE level 5 (in purple), level 6 (in yellow), and level 7 (in red) rapidly expand to x = 400–500 m after 9 min, indicating that the risk of serious injury and death after 10 min is great in the area near the fire point.

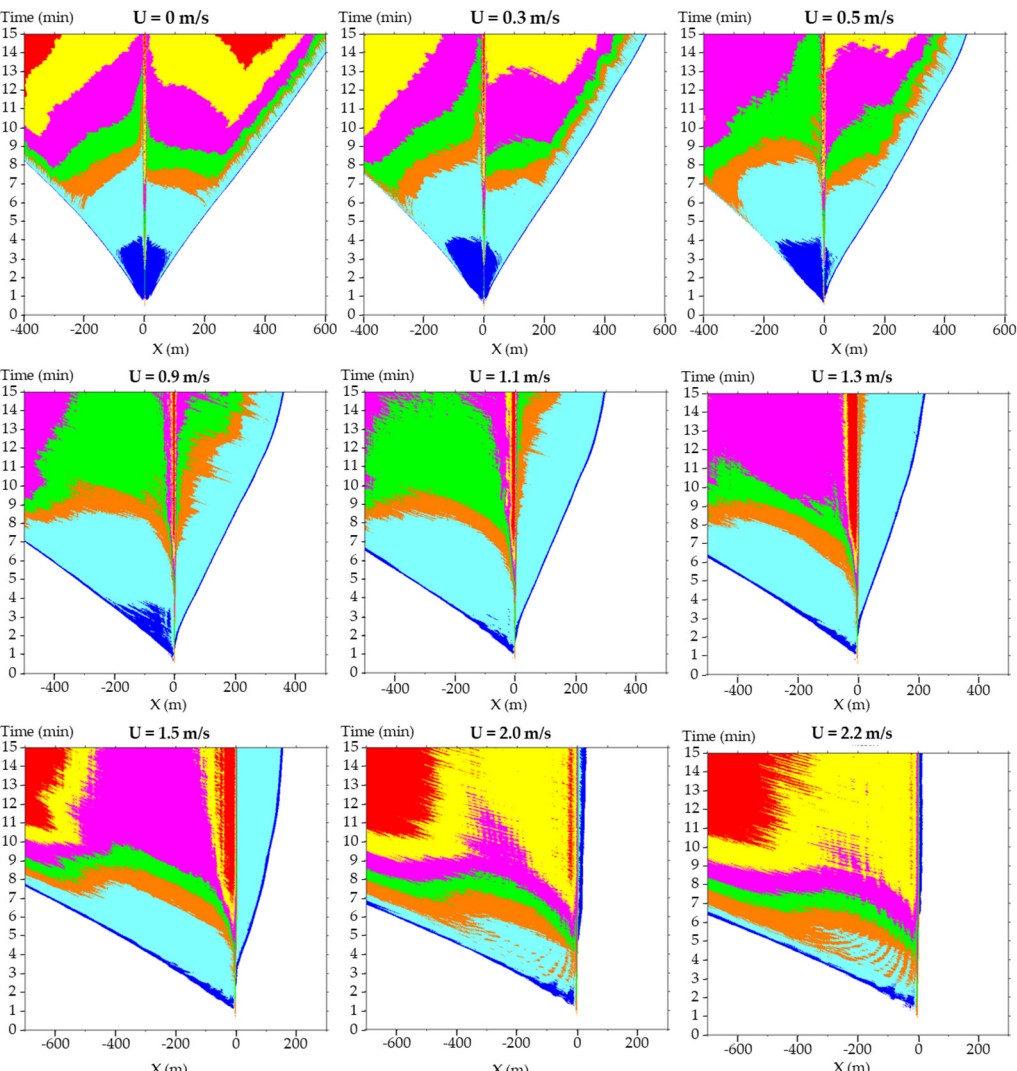

**Figure 5.** SE map in cases of Um = 0–2.2 m/s (rectangular tunnel).

Next, the evacuation conditions downstream of the fire are discussed, the SE level worsens significantly when the upstream back-layering disappears. The longitudinal velocity exceeds 1.5 m/s and is lower than 0.5 m/s, revealing a relatively worse smoke environment, with the appearance of high SE levels (levels 6–7). When U = 1.1 m/s, the downstream side can maintain SE level 2 (in light blue) for around 6–7 min, but smoke gradually diffuses to the road surface, causing the smoke environment (SE) to change to level 3 (in orange). When U = 0.9 m/s, the downstream side can maintain SE level 4 (in green) in around 9–10 min; this indicates the risk of minor injuries when evacuees delay egressing until after 9–10 min. When U = 0.5 m/s, the smoke environment deteriorates with the appearance of SE level 5 (in purple). SE level 6 (in yellow) appears after 10 min, when U = 0 m/s and 0.3 m/s. In the case of U = 0 m/s, SE level 7 (in red) occurs at the location of x = −400 m at t = 12 min and then spreads to x = −300−−400 m in 15 min due to the smoke descending. In contrast, SE level 6 (in yellow) and SE level 7 (in red) appear after 10 min in the case of U = 1.5–2.2 m/s due to the strong forced ventilation, resulting in significant smoke diffusion rather than the smoke descending.

Figure 6 shows the SE map of the horseshoe-shaped tunnels. Compared with the SE map of the rectangular tunnel, back-layering exists, and the back-layering length is greater than the rectangular tunnel when U is greater than 2.2 m/s. Clearly, the critical velocity increases with the increase in the cross-sectional areas of tunnels, but it decreases with the

increase in tunnel height (under the condition of the same width), which is also indicated by Li and Ingason (2017) [47].

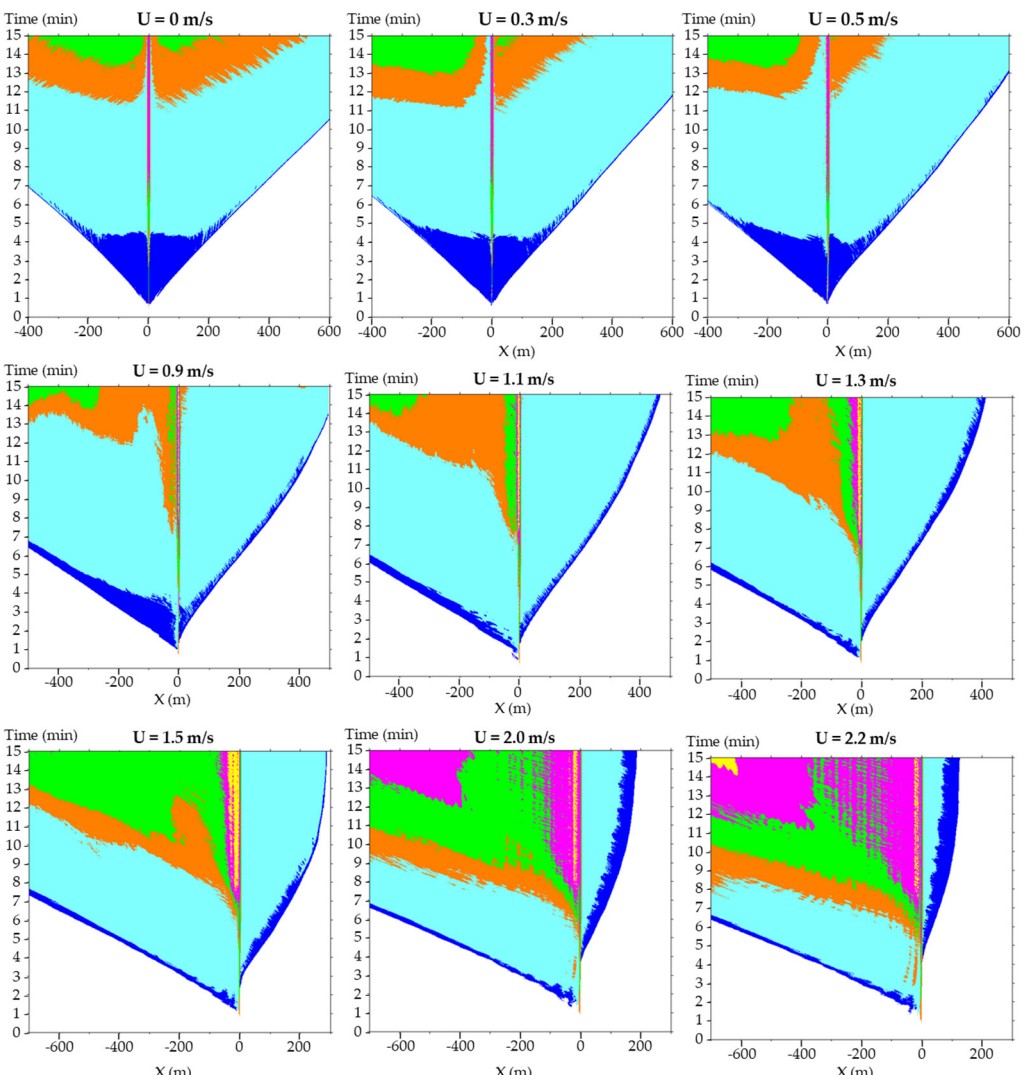

**Figure 6.** SE map in cases of U = 0.3–2.2 m/s (horseshoe shape).

The upstream of the fire maintains an environment with SE level 2 (in water blue) or lower when U > 1.1 m/s. SE level 3 (in orange) appears upstream when U < 0.9 m/s. Nevertheless, no worse conditions than SE level 5 occur upstream under current longitudinal velocity conditions.

When focusing on the downstream of the fire in Figure 6, the smoke environment maintains SE level 2 (in water blue) over 10 min in cases where U < 0.9 m/s; however, SE level 3 (in orange) gradually appears on the downstream side at around t = 12 min. This indicates significant smoke descending after 10 min when U = 0–0.5 m/s. In the case of U = 0.9 m/s, SE level 4 (in green) occurs on the downstream side at t = 14 min, occurring early with the longitudinal velocity decrease. However, in cases of U = 1.5–2.2 m/s, the occurrence of SE level 4 is also early, around t = 9–10 min. SE level 4 occurs when U = 1.5–2.2 m/s because the strong forced ventilation results in significant smoke diffusion rather than the smoke descending phenomenon that appears in low-velocity cases (U < 0.9 m/s). In the case of U > 2.0 m/s, SE level 5 (in purple) appears and gradually spreads. Despite this, no significant SE level 6 (in yellow) and SE level 7 (in red) appear when U= 1.5–2.2 m/s.

Comparing the rectangle-shaped and horseshoe-shaped tunnels, significantly worse SE levels occur in the rectangular tunnels due to relatively low tunnel height. The relatively

high tunnel height of the horseshoe-shaped tunnels is advantageous to keep the smoke in place at a higher level.

Moreover, in the rectangular tunnel, compared with other velocity conditions, 0.9–1.1 m/s reveals a less serious SE level distribution for both the upstream and downstream. In the horseshoe-shaped tunnel, a less serious SE level distribution for both the upstream and downstream appears for velocity conditions of 0.3–0.9 m/s. We consider that the difference in velocity for maintaining a less serious SE level in a horizontal rectangular tunnel and a horizontal horse-shaped tunnel is due to the influence of the cross-section area and disturbance from longitudinal ventilation.

In the horizontal horseshoe-shaped tunnel, a relatively large cross-section contributes to maintaining the stratified smoke in the tunnel ceiling, without it descending. Relatively low longitudinal ventilation would not affect this stratified state of smoke. In the rectangular tunnel, the small cross-section cannot maintain the stratified smoke in the tunnel ceiling without it descending as a result of low longitudinal ventilation conditions. Moreover, the strong forced ventilation also resulted in significant smoke diffusion toward to the road surface in the case of U = 1.5–2.2 m/s. Balancing the influence of the cross-section areas and disturbance from longitudinal ventilation, a less serious SE level occurred in the case of a velocity of 0.9–1.1 m/s both upstream and downstream of the fire.

The widely adopted ventilation mode (target velocity U = 1 m/s) for congested traffic conditions in European countries [48–50] is similar to the results for the rectangular tunnel. The widely adopted zero flow ventilation mode (target velocity U = 0 m/s) for congested traffic conditions in Japan [51] is similar to the results for the horseshoe-shaped tunnel. Thus, it is evident that the suitable velocity for maintaining an acceptable SE level varies depending on the tunnel geometry.

In addition, the main objective of tunnel fire strategy in Japan is that the tunnel users can be evacuated safely within 10 min during fire incidents. Although the present study mainly considered the interaction between velocities and smoke distribution instead of further including the influence of vehicles on smoke distribution, according to the simulation results, the strategy that the applies the zero flow ventilation mode for the objective of completing evacuation in 10 min seems possible in the horizontal horseshoe-shaped tunnel. However, according to the results in Figure 5, such strategies should be carefully considered when applied in rectangular tunnels because the relatively low height of the rectangle makes smoke descend faster, and SE level 5 appears upstream and downstream earlier than the 10 min thresholds.

*4.2. SE Map with the Effect of Longitudinal Gradients*

In this subsection, we further analyze the effect of longitudinal gradients (G, %) on the SE map in the rectangular and horseshoe-shaped tunnels.

Figure 7 illustrates the SE map of the rectangular tunnel in cases of U = 0, 0.5, and 1.1 m/s with G = 0, 2, and 4%. The displayed total longitudinal region is 1300 m in each case. In Figure 7, the right side of the fire is defined as upstream, and left side of the fire is defined as downstream.

When U = 0 m/s with G = 0%, SE level 4 (in green) first appears at x = −200 and 200 m at t= 7 min, SE level 5 (in purple) appears at x = −300 and 300 m at t = 8 min, SE level 6 (in yellow) first appears at x = −300 and 300 m at t = 10 min, and SE level 7 (in red) first appears at x = −300 m at t = 12 min. Hence, it can be read that the smoke layer propagates horizontally and then descends at a location far from the fire source in the case of G = 0%. In contrast, in the cases of G = 2% and 4%, the SE level range is 4–7, which shows that descending smoke is moved upstream. SE levels 6 and 7 occur earlier (at around 7–8 min) and gradually spread to x = −100–700 m at t = 15 min. In addition, the range experiencing the hazard of SE levels 6 and 7 for G = 4% is wider than for G = 2%.

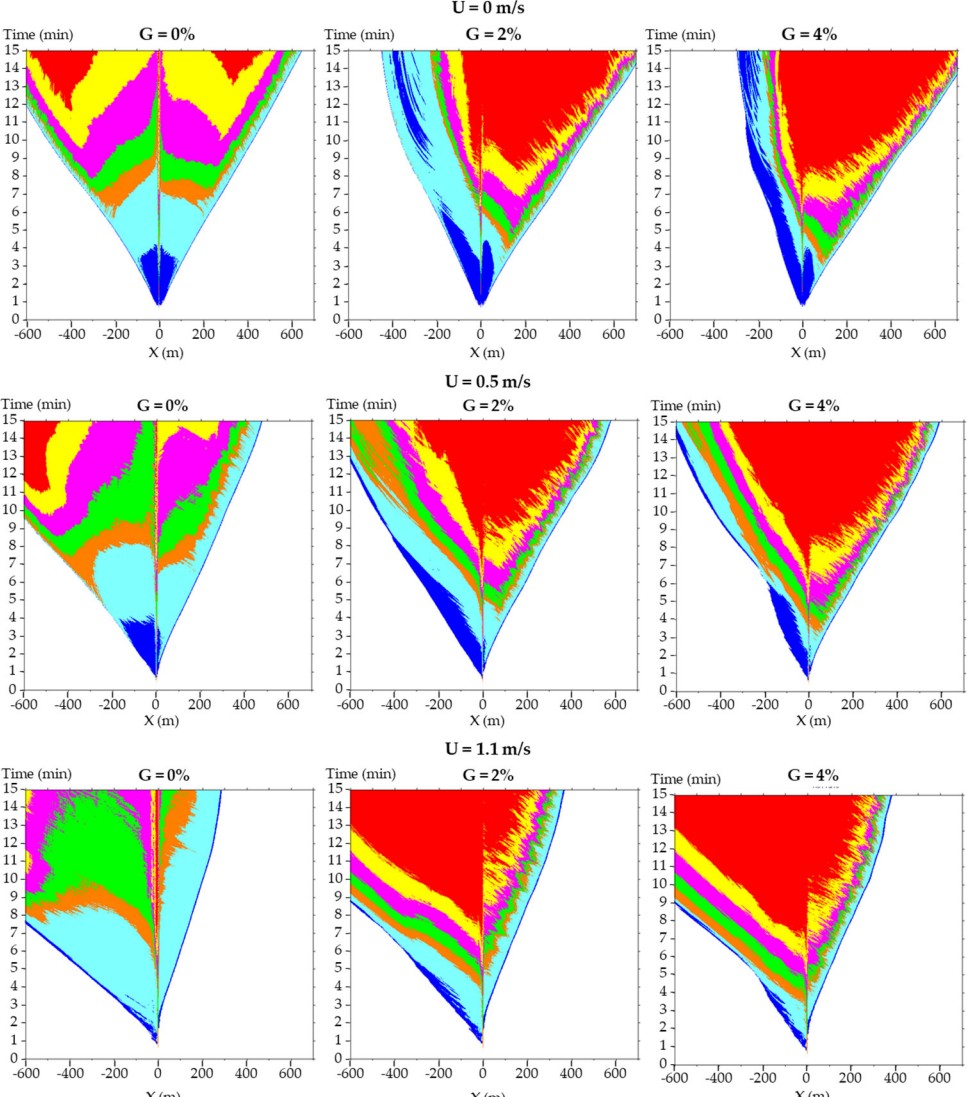

**Figure 7.** SE map of rectangular tunnel with change in gradient.

When U = 0.5 m/s, it is observed that the range of each SE level is moved upstream in cases where G = 2% and 4%. The area suffering the hazards of SE levels 1–3 is larger than that in the case of U = 0 m/s with G = 2% and 4%; however, the area suffering the hazards of SE levels 4–7 is similar to the case of U = 0 m/s with G = 2% and 4%.

When U = 1.1 m/s, the SE level is significantly worse in the case of G = 2% and 4% than in the case of G = 0%. Moreover, compared with G = 2%, SE levels 6–7 are moved upstream when G = 4%. Thus, it is confirmed that when the gradient increases to 4%, the smoke descending phenomenon moves upstream, the same as in the case of U = 0 m/s and 0.5 m/s. The smoke descending phenomenon is worsened in the cases with gradients than in the cases without gradients.

Comparing the cases with and without gradients in the rectangular tunnel, we find that the velocity conditions with relatively less severe SE levels for both the upstream and downstream sides were 0.9–1.1 m/s in cases without gradients. However, this velocity range should be considered to decrease in the same tunnel type with inclination, because the SE levels 6–7 (representing smoke descending) occur in the inclined rectangular tunnel in the case of U = 1.1 m/s.

Figure 8 illustrates the SE map of the horseshoe-shaped tunnel with G = 0, 2, and 4%. The displayed total longitudinal region is 1300 m in each case.

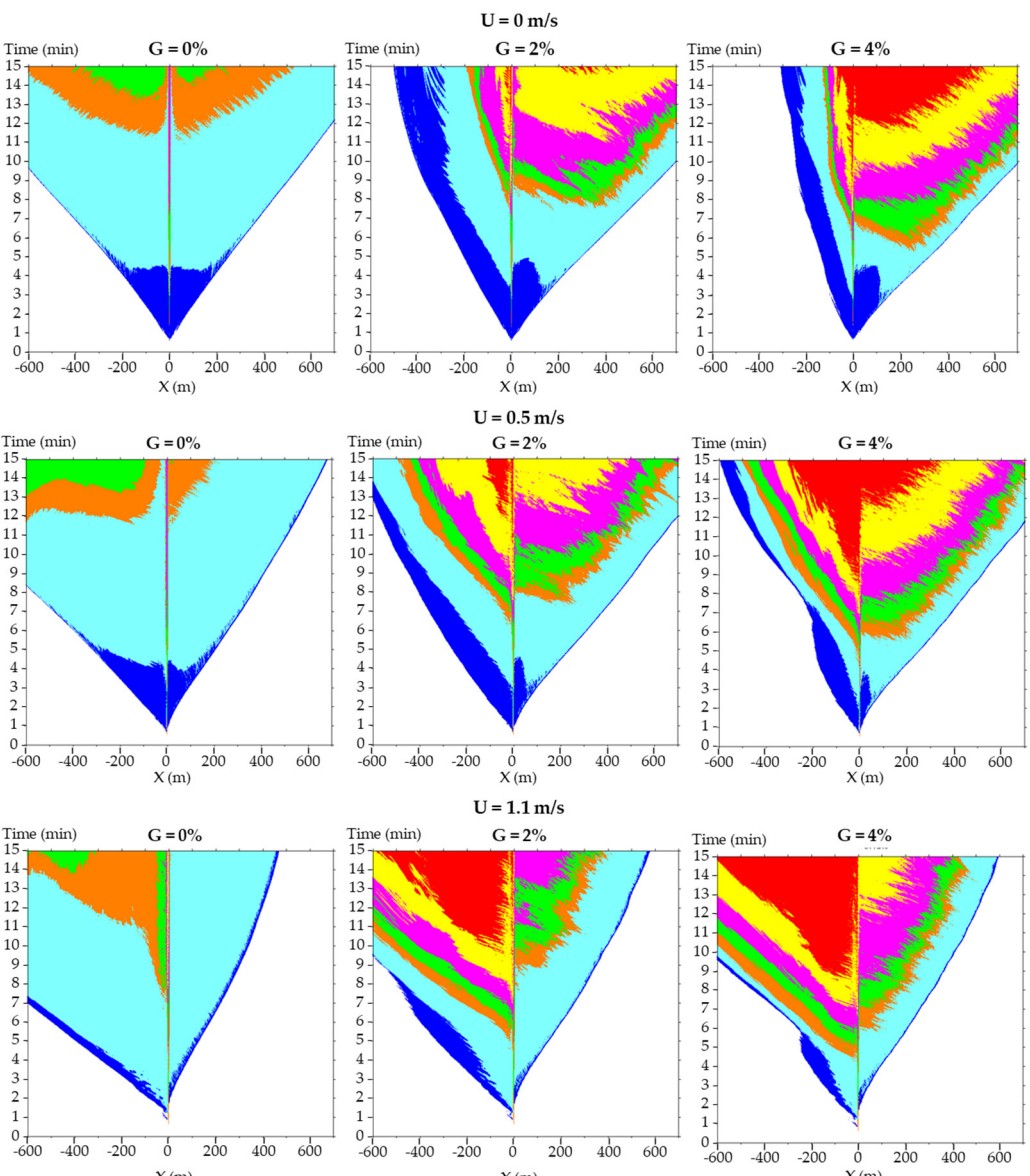

**Figure 8.** SE map of horseshoe-shaped tunnel with change in gradients.

When U = 0 m/s with G = 0%, the highest SE level was 4 (in green) in 15 min, while the highest SE level reached was 6 (in yellow) upstream in the case of G = 2% and 7 (in red) upstream in the case of G = 4%. Moreover, SE level 4 (in green) first appeared at t = 9 min for G = 2% but at t = 8 min for G = 4%; SE level 6 (in yellow) first appeared at t = 11 min for G = 2% but at t = 10 min for G = 4%. It is clear that smoke descending on the upstream side is more significant with the gradient increase.

When U = 0.5 m/s, it is observed that SE levels 3–7 spread both upstream and downstream in the cases of G = 2 and 4%; however, the range of SE levels 5–7 is wider in cases of G = 4% than in G = 2%. SE level 6 (in yellow) spreads to x = −300–400 m in G = 2% but spreads to x = −300–500 m in G = 4%; SE level 7 (in red) spreads to x = −100–0 m in G = 2% but spreads to x = −200–300 m in G = 4%.

When U = 1.1 m/s, the range of SE levels 3–5 spreading upstream increases with the gradient increase to G = 4%. SE levels 6 and 7 did not occur upstream in the case of G = 2% but SE level 6 appears when the gradient increases to G = 4%. Hence, concerning higher gradients, the smoke descent of SE levels 6 and 7 (shown as yellow and red areas in Figure 8) commences earlier and spreads wider both upstream and downstream.

For the rectangular tunnel, the smoke descending phenomenon is worse in the cases with gradients than in the cases without gradients. SE level 3 (in orange), which represents the acceptable safety criteria widely adopted in Japan, is achieved in under 10 min in all cases with gradients, as seen in Figure 8.

A comparison of the cases with and without gradients in the horseshoe-shaped tunnel also shows that the velocity conditions with relatively less serious SE levels for both the upstream and downstream are 0.3–0.5 m/s in cases without gradients; however, this velocity range should be considered to be lower in the inclined tunnel, since a relatively less serious SE level distribution is found in the case of U = 0 m/s with gradients.

In addition, the SE level assessment in both rectangular and horseshoe-shaped tunnels reveal an obvious rise between 10 and 15 min. This might be because those who cannot evacuate the tunnel in 10 min, such as the elderly or people with disabilities, would face a higher risk of injury or death.

The number of fatalities due to 15 min of smoke exposure was not investigated in the present paper, and this point is to be clarified in further research.

## 5. Conclusions

In this study, we improved the existing tunnel risk assessment approach by proposing an assessment based on the influence of visibility and toxic gas on evacuation. The assessment method of smoke hazards that integrates the factors of toxic gases (CO and HCN) and visibility was considered. The assessment method proposed by this study quantifies the environmental risk of the smoke environment, including toxicity and visibility, and shows the risks using a 2D map to clarify the possible hazards faced during evacuation. For demonstrating the applicability of the proposed method, an ultra-fast-growing bus fire of 30 MW was designed to analyze the influence of longitudinal velocities, cross-section types, and gradients on smoke distribution using CFD simulation. The main conclusions of this study are summarized as follows:

- Because the horseshoe-shaped tunnel has a relatively large cross-section, the range where the smoke layer descended to affect evacuees (SE levels 4, 5, 6, and 7) is smaller than that of a rectangular tunnel, even at different longitudinal velocities and gradient conditions.
- In the analysis of the SE level in different cross-section types and longitudinal velocities under the condition of no vehicle, the velocity of around 0.9–1.1 m/s can maintain a relatively less serious SE level both upstream and downstream in a horizontal rectangular tunnel. A velocity of around 0.3–0.5 m/s can maintain a relatively less serious SE level both upstream and downstream in a horizontal horseshoe-shaped tunnel.
- SE level assessment in both rectangular and horseshoe-shaped tunnels reveal an obvious increase within 10–15 min. This might be because those who could not evacuate the tunnel in 10 min, such as the elderly or people with disabilities, would face a higher risk of injury or death.
- In the case of an inclined tunnel, it can be found that for tunnels that are not rectangular or horseshoe-shaped tunnels, the SE level near the fire source is significantly deteriorated. The longitudinal velocity range for maintaining a relatively less serious SE level is slightly reduced compared with horizontal tunnels.
- The usage of grading and a graphical approach to illustrate the risk of smoke distribution and toxic gas exposure in this study allows more comprehensive estimation of the threats in the tunnel region and the degree of possible harm to the evacuees.

The findings of this study contribute to a more comprehensive risk evaluation of smoke distribution by considering visibility and toxic gas and provide useful insights for the emergency operation of the longitudinal ventilation system in the event of tunnel fires. However, the influence of the existence of vehicles on the SE level requires further study. It remains a future task to evaluate the SE level in the cases with vehicles and combine the evacuation models for a comprehensive quantitative risk analysis for tunnel fire safety.

**Author Contributions:** Conceptualization, H.-R.H., H.-C.C. and N.K.; methodology, H.-R.H., H.-C.C. and N.K.; software, N.K.; validation, H.-R.H., H.-C.C. and N.K.; formal analysis, H.-R.H., H.-C.C. and N.K.; investigation, H.-R.H., H.-C.C. and N.K.; resources, N.K.; data curation, H.-R.H. and H.-C.C.; writing—original draft preparation, H.-R.H. and H.-C.C.; writing—review and editing, N.K., M.S. and M.H.; visualization, H.-R.H. and H.-C.C.; supervision, N.K., M.S., M.H., S.-W.C. and T.-S.S.; project administration, N.K., M.S., M.H., S.-W.C. and T.-S.S.; funding acquisition, N.K., M.S., M.H., S.-W.C. and T.-S.S. All authors have read and agreed to the published version of the manuscript.

**Funding:** Grants were received from the Ministry of Education, Culture, Sports, Science and Technology, Japan in aid of scientific research (number: 20K05008).

**Institutional Review Board Statement:** Not applicable.

**Informed Consent Statement:** Not applicable.

**Data Availability Statement:** Not applicable.

**Acknowledgments:** The authors are sincerely grateful for the funding aid from the Ministry of Education, Culture, Sports, Science and Technology, Japan.

**Conflicts of Interest:** The authors declare no conflict of interest.

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
