# Peer review of "Assessment Method Integrating Visibility and Toxic Gas for Road Tunnel Fires Using 2D Maps for Identifying Risks in the Smoke Environment"

_fire, doi:10.3390/fire6040173_

Round 1

Reviewer 1 Report

This paper proposes a quantitative method for assessing the risk of the smoke environment in road tunnel fires based on a numerical simulation. The proposed method was further applied to investigate the risk of tunnel fire with various tunnel cross-section types, longitudinal velocities, and gradients. This is interesting, but there are some small issues that need to be clarified.

1The horizontal tunnel is 1300m in length and the length of inclined tunnel is 1800m. What is the basis for setting the tunnel length? Why not set the same length

2Where is the fire source in the longitudinal direction of the tunnel? It is said that A fire source is put in the center of the tunnel, but from Figure 5 (Um>1.5m/s), there is at least 700m upstream of the fire source, and the total length of the horizontal tunnel is 1300m. Obviously, the fire source is not in the middle of the longitudinal direction of the tunnel.

3The velocity of around 0.9-1.1 m/s can maintain a less serious SE level both up- stream and downstream in a horizontal rectangular tunnel, and around 0.3-0.5 m/s in a horizontal horse-shoe-shaped tunnel. The results of different cross sections differ so much. Can you further explain the reason for this difference

Reviewer 2 Report

The paper presents an effective visualization of tunnel smoke filling process that combines visibility and FED. Novelty is medium level, although the method can be very useful practical risk analysis projects. Biggest problems are the unclear justification of input values, lack of sensitivity study for the chosen inputs, and the confusing use of FED approach. I would like to see a verification of the direct FED integration from a selected place and comparison with SE map - is the result the same?

Detailed comments:

- Grid sensitivity study should consider a wider range of different grid sizes. Not the cell sizes differ less than 30 %. And still the results indicate grid dependency. Conclusion of grid-independence is not convincing.

- I accept the fact of neglecting radiation for the sake of computational savings, but it is wrong to say it would be difficult to reproduce in simulations. Radiation solvers have been a standard part of CFD codes more than 20 years.

- What do you mean by 'smoke generation' in Fig 3? Soot production? It seems this corresponds to a soot yield of 0.12 (0.09 kg/s /(30 MW/40 MJ/kg)), but this has not been explained.

- How is the fire, i.e. energy and mass source actually modelled? 

- Extinction coefficient has a unit (1/m), and thus the coefficient 10 in eq (3) also has a unit (m2/g). Taking a logarith of a dimensional quantity (M) in the second part of Eq. (3) is not mathematically ok. Please fix.

- Saying 'CO and HCN generation rate is assumed proportional to fire growth rate' is wrong. I believe you mean 'proportional to HRR'.

- What are the CO and HCN yields? It is not a good practice simple give total yields as it is difficult for readers to relate these values to the toxic nature of the fire.

- I am not sure I understand how the SE levels are determined from the CFD results. This part needs to be clarified. Why do you introduce the concept of STV, and don't use the FED as such?

- English language needs to be corrected. There are numerous grammatic or spelling errors, so many that I cannot list them. For example, the use of 'radiant' as a noun.

- Time-related results are specific to the chosen fire growth rate (which is not discussed but seems to be ultra-fast growth rate). This should be mentioned in conclusions. 

Round 2

Reviewer 2 Report

Thank you for carefully considering the comments